# Effects of Destruxin A on Silkworm’s Immunophilins

**DOI:** 10.3390/toxins11060349

**Published:** 2019-06-18

**Authors:** Jingjing Wang, Qunfang Weng, Qiongbo Hu

**Affiliations:** Key Laboratory of Natural Pesticide & Chemical Biology, Ministry of Education; College of Agriculture, South China Agricultural University, Guangzhou 510642, China; wangjingjing@stu.scau.edu.cn (J.W.); wengweng@scau.edu.cn (Q.W.)

**Keywords:** Destruxin A, peptidyl-prolyl isomerase, FK506 binding protein, cyclosporine A, FK506, *Bombyx mori*

## Abstract

Destruxin A (DA), a major secondary metabolite of *Metarhizium anisopliae*, has anti-immunity to insects. However, the detailed mechanism and its interactions with target proteins are elusive. Previously, three immunophilins, peptidyl–prolyl cis–trans isomerase (BmPPI), FK506 binding-protein 45 (BmFKBP45) and BmFKBP59 homologue, were isolated from the silkworm, *Bombyx mori* Bm12 cell line following treatment with DA, which suggested that these proteins were possible DA-binding proteins. To validate the interaction between DA and the three immunophilins, we performed bio-layer interferometry (BLI) assay, and the results showed that DA has interaction with BmPPI, whose affinity constant value is 1.98 × 10^−3^ M and which has no affinity with FKBP45 and FKBP59 homologue in vitro. Furthermore, we investigated the affinity between DA and human PPI protein (HsPPIA) and the affinity constant (K_D_) value is 2.22 × 10^−3^ M. Additionally, we compared the effects of silkworm and human PPI proteins produced by DA and immunosuppressants, cyclosporine A (CsA), and tacrolimus (FK506), by employing I2H (insect two-hybrid) in the SF-9 cell line. The results indicated that in silkworm, the effects created by DA and CsA were stronger than FK506. Furthermore, the effects created by DA in silkworm were stronger than those in humans. This study will offer new thinking to elucidate the molecular mechanism of DA in the immunity system of insects.

## 1. Introduction

Destruxins are cyclodepsipeptidic mycotoxins secreted by the entomopathogenic fungus *Metarhizium anisopliae*; they have 39 analogues and play a pivotal role in the pathogenicity of *M. anisopliae* against insect hosts [1,2]. Destruxin A (DA) (Figure 1A), the major destruxin analogue, displays diverse bioactivities, including insecticidal, antifeedant, and growth-retardant effects with inhibition of immunity to insects, and anticancer activity in humans [3,4]. Therefore, DA is considered a candidate medical and insecticidal drug. However, the molecular mechanisms of DA bioactivities have not been elucidated, although many research results have been published [5]. It was reported that DA damages insect tissues such as the midgut, visceral muscles, Malpighian tubules, and hemolymph [6,7]. Moreover, DA is a type of cationic ionophore and V-type ATPase inhibitor, which can change the homeostasis of intracellular Ca^2+^ and H^+^ [8,9]. In particular, DA has attracted attention for its immunosuppressant activity. It has been reported that DA impacts antimicrobial peptides in *Drosophila melanogaster* [10], regulates the gene expression of BmRelish and BmRel of the Imd pathway in *Bombyx mori* [11], and influences the expression of immunity-related genes in insects [12,13,14]. However, there has been no significant progress on clarifying the molecular toxicology of DA. 

In order to identify the target protein of DA, we performed experiments using drug affinity responsive target stability (DARTS) [15] and identified dozens of possible DA-binding-proteins in *B. mori* Bm12 cells (not published). As described above, several reports showed that DA acts in the immune-related pathway as an inhibitor. Therefore, we selected all immunophilins in DARTS experiments, peptidyl–prolyl cis–trans isomerase (BmPPI), FK506 binding protein 45 (BmFKBP45) and BmFKBP59 homologue, to verify the affinity with DA. Structurally, BmPPI belongs to the cyclophilin superfamily, which means it has higher affinity with cyclosporine A (CsA) [16], a cyclo-peptide immunosuppressant used for the therapy and prophylaxis of graft rejection in human organ transplantation (Figure 1C) [17]; BmFKBP45 and BmFKBP59 homologue were classified in the FKBP super family, which means they have higher affinity with FK506 (tacrolimus, Figure 1B), another polyketidic immunosuppressant produced by *Streptomyces tsukubaensis* [18]. In this research, we investigated the affinity between three immunophilins with DA in vitro by bio-layer interferometry (BLI) [19]. Moreover, we aimed to use an insect two-hybrid (I2H) system [20] to compare the effects of DA, CsA, and FK506 on PPI proteins of silkworm BmPPI and human PPI protein (HsPPIA). This study provides new insights to better understand the molecular toxicology of DA and its potential applications.

## 2. Results

### 2.1. The Affinity of DA with BmPPI, BmFKBP45, BmFKBP59 Homologue and HsPPIA

Heterologously recombinant proteins were expressed and purified from a prokaryocyte expression system in *Escherichia coli* (Appendix A). In BLI assay, proteins were labeled to Ni–NTA biosensors, and data were collected and analyzed after serial dilutions of DA flowed through sensors. The BLI results indicated that DA and BmPPI show slight affinity with an affinity constant (K_D_) value of 1.98 × 10^−3^ M (Figure 2, Table 1). Although the K_D_ value is a little large, BmPPI is a DA-binding protein is no doubt. However, the results also showed that there are no interactions in DA with BmFKBP45 and BmFKBP59 homologue, because they have so large values of K_D_, K_ON_ and K_DIS_ (Figure 2, Table 1). Additionally, we assessed the affinity between DA with HsPPIA, and the results exhibited that the K_D_ value is 2.22 × 10^−3^ M. Interestingly, the same affinity level of DA with BmPPI and HsPPIA revealed that DA might interact with some domains in PPI structure. Unfortunately, both affinities were relatively weak at 10^−3^ level.

### 2.2. Influences of DA, CsA, and FK506 on the Interactions of BmPPI–Bmo.3174 and HsPPIA–PPP3CA

The insect two-hybrid (I2H) system in the *Spodoptera frugiperda* Sf-9 cell line was employed to compare the influences of DA, CsA, and FK506 on PPI proteins of silkworm’s BmPPI and humans’ HsPPIA (Figure 3A). We respectively selected the BmPPI and HsPPIA interacting protein, Bmo.3174 and PPP3CA from the String Protein–Protein Interaction Database. The interactions of BmPPI–Bmo.3714 and HsPPIA–PPP3CA were proven through I2H. The lumen values of the Sf-9 cells co-transinfected by PIE–BmPPI–AD, PIE–Bmo.3174–DBD, and PIE–Luc vectors, or by PIE–HsPPI–DBD, PIE–PPP3CA–AD, and PIE–Luc were respectively recorded as 580 and 5500, which is apparently higher than 40 of CK (Sf9 cell) (Figure 3B).

The interferences caused by DA, FK506, and CsA in silkworm’s BmPPI–Bmo.3714 and humans’ HsPPIA–PPP3CA were examined through the relative luciferase activity (RLA) which divides the drug group (0.02, 0.2 and 2 mg/L) by the control group (DMSO) (Figure 3C). Apparently, DA and CsA, in a dose-dependent manner, influenced the protein interactions, but FK506 seemed to have a weaker effect on the proteins. In BmPPI–Bmo.3714 cells, the interferences caused by DA and CsA were significantly higher than that by FK506, which suggested that the interaction of DA and CsA with BmPPI is stronger than that of FK506. It was also indicated that DA at 0.2 mg/L dose had stronger effects on protein interactions, compared with CsA and FK506. Correspondingly, in HsPPIA–PPP3CA cells, the interferences created by CsA were higher than those by DA and FK506 at all treatments. Obviously, in all concentrations, the interferences generated by DA in silkworm were stronger than human. 

According to both results, there are reasons to explain these experimental phenomena. In *B. mori*, the effects created by DA and CsA were stronger than those by FK506, and this confirmed the data from Section 2.1 that DA only has affinity with BmPPI but not FKBPs. Structurally, it also makes sense, as DA and CsA are cyclopeptide and fungal metabolites, whereas FK506 is a macrolide originating from streptomycete. Moreover, on account of DA being secreted from entomopathogenic fungus, the effects generated by DA in silkworm were stronger than those in mammal humans.

## 3. Discussion

In order to elucidate the action mechanism of DA in molecular toxicology, we performed DARTS experiments, and three immunophilins were identified as possible DA-binding proteins. However, affinity assays of BLI revealed that only BmPPI binds to DA and FKBPs. Although in DARTS, theoretically, all isolated proteins were DA-binding proteins, there were false positives, such as higher expressed or induced proteins under stress or those binding proteins and so on. Obviously, FKBPs were false positive after affinity assays even though FKBP45 appeared in two treatments. In previous studies, FKBPs were reported to aid protein folding after stress in variant species. Therefore, we performed RT-qPCR to detect the gene expressions of FKBPs under DA stress, and the results showed that they were upregulated after DA treatments at 6 and 12 but not 24 or 48 h (no published). It was confirmed that FKBPs were induced by DA, which is in accord with in our previous studies, heat shock proteins (HSPs) were induced, and one of these binds to DA [22]. Interestingly, other reports revealed that HSPs can interact with FKBPs [23,24], and it might be suggested that FKBPs were bound to HSPs and other proteins and appeared in our DARTS experiments. 

DA has been shown to have substantial insecticidal and anti-immunity activities in insect pests [10]. However, it remains unclear which immune-related proteins have more affinity to DA. The present study demonstrated that BmPPI protein of *B. mori* is a binding protein of DA. However, the affinity value is weak, with an affinity constant value K_D_ value of 10^−3^ M in the BLI test, which seems to make it difficult to draw a conclusion that DA is an immunosuppressant in insects which targets BmPPI. However, DA in a dose-dependent manner influences the interaction of BmPPI and Bmo.3174 proteins, and the lowest affective dose of DA is 0.2 mg/L (0.35 µM), which suggests DA has better affinity to BmPPI protein in vitro. Obviously, from BLI and I2H results, it seems that in a molecular mechanism, DA was more similar with CsA than FK506. It was more likely that DA acts as an immunosuppressant in insects in the CsA-related pathway with its innate immunity. Bmo.3174 is predicted as a *B. mori* cell division cycle 5-like protein (BmCDC5). CDC5 is a core component of the putative E3 ubiquitin ligase complex, a highly conserved protein in eukaryotes [25,26]; BmCDC5 has a 65% identity with humans’ CDC5. This complex has been shown to have a role in pre-messenger RNA splicing from yeast to humans. Previously, we found that DA influences the splicing of pre-mRNA of silkworm’s immune-related gene, BmRelish, and results in expression of BmRelish1 which is higher than that of BmRelish2 [11]. Therefore, we hypothesized that DA might interfere with the interaction of BmPPI with Bmo.3174 and lead to irregularity splicing of the downstream immune-related gene in the immune-related pathway. Further, our experimental results illustrated that DA has similar affinity with silkworm and human PPI proteins, although BmPPI and HsPPIA have ~50% homologous gene sequences. However, from the I2H results, DA was shown to have better toxicity in entomogenous fungus *M. anisopliae* host than in human. 

For our experiment, we employed an in vitro method termed BLI [19] that directly detects affinity between a drug and recombinant protein. We used the I2H system [20,21] for in vivo verification; this two-hybrid system was suitable for insect research and has great benefits, including simplicity, ease of operation, and high sensitivity. Moreover, we chose I2H for biological verification in vivo for several reasons: The proteins in vivo are involved in an interaction network, its conformation change depended on which (protein) ligand interacted with it, and the reaction environment and solution were living cells and culture medium.

## 4. Conclusions

In conclusion, the present study showed that DA displayed affinity interaction with the protein PPI in the model insect *B. mori* and human but not with FKBPs. Additionally, our results also suggest that in a mechanism, DA is more similar with CsA than FK506, and the effects of PPIs are stronger in *B. mori* than in humans. Overall, this study provides baseline data on the binding affinity of DA against target pests and human proteins and provides a platform for developing novel pesticides as well as human drugs.

## 5. Materials and Methods 

### 5.1. Cell Lines, Culture, and Treatment

The *Spodoptera frugiperda* 9 (Sf-9) cell line was cultured in SFX culture medium (HycloneTM, Pittsburgh, MA, USA) with 5% fetal bovine serum (GibcoTM Waltham, MA, USA). The Sf9 cell line was maintained at 27 °C and passaged over a period of 2–4 days. The cell line in logarithmic phase was used for the experiment. DA was isolated and purified from the *Metarhizium anisopliae* var. anisopliae strain MaQ10 in a laboratory [27]. CsA and FK506 were purchased from MedChem Express. DA, CsA, and FK506 were stored in −20 °C as powder and dissolved using dimethyl sulfoxide (DMSO) to working concentration in experiment. The control group was only a supplement with the same volume of DMSO.

### 5.2. Bio-Layer Interferometry (BLI)

All proteins were prepared by expression in *E. coli*. These were tagged with His-tag and purified using nickel affinity chromatography (S1). BLI analysis was performed on a ForteBio Octet QK System (K2, Pall Fortebio Corp, Menlo Park, CA, USA). Generally, the protein samples were coupled with a biosensor for immobilization. Serial gradient dilutions of DA (500, 250, 125, 62.5, 31.25, 15.63, and 7.813 µM) were used for treatment. PBST buffer (0.05% Tween20, 5% DMSO) was used for the reference and dilution buffers. The working procedure was baseline for 60 s, association for 60 s, and dissociation for 60 s. Finally, the raw data were collected and processed with Data Analysis Software (9.0, Pall Fortebio Corp, Menlo Park, CA, USA).

### 5.3. Insect Two-Hybrid (I2H)

Insect two-hybrid (I2H) [20] was used to measure protein interaction of PPI protein with its interacting protein under DA, CsA, and FK506 interference. The nucleotide sequences of BmPPI (NM_001047055.2) and HsPPIA (NM_021130.4), and their interaction proteins Bmo.3174 (XM_012694714.1) and HsPPP3CA (NM_000944.4) were acquired from NCBI. A gateway system (Invitrogen) was employed to construct entry vectors with a pENTR/D-TOPO vector, and fresh PCR product which amplified with a 5′ specific primer. The destination vectors, PIE–AD, PIE–DBD, and PIE–Luc, were a gift from Takahiro Kusakabe. For recombination, based on the user guide, LR reaction was applied to constructed I2H vectors PIE–BmPPI–AD, PIE–Bmo.3174–DBD, PIE–HsPPIA–DBD, and PIE–PPP3CA–AD.

The abovementioned constructed I2H vectors were co-transfected into Sf9 cell line. Lipofectimine2000 (Thermo Fisher, Shanghai, China) and plasmids were diluted with the serum-free SFX culture medium and incubated at room temperature for 5 min. Then, the two solutions were mixed and incubated at room temperature for 30 min to form a plasmid–lipofectimine2000 mixture for the cell transfection experiment. For cell transfection, the Sf9 cells in the logarithmic phase were moved into 12-well plates and cultured with an SFX culture medium. After 24 h, the culture medium was sucked out and supplemented with 100 µL of serum-free medium, which was then gently shaken. Then, 400 μL of plasmid–lipofectimine2000 mixture was added to each well and incubated at 27 °C. After 24 h, the supernatant was washed out, and transfection was terminated. Then, fresh SFX medium-containing serum was added, and incubation occurred for 24 h for the combination treatment with drugs and a survey of interference of interaction. The experiments were replicated three times. The control group was only treated with DMSO. After the cell extracts were co-transfected in the Sf9 cell line and incubated for 48 h with drugs, the luciferase activities in them were determined using a luciferase reporter assay system (Promega, Beijing, China) and Synergy™ H1 (BioTek, Winooski, VT, USA). Data were collected and analyzed. The means and DMRT (Duncan’s multiple range test) were evaluated by employing SPSS software (IBM, Armonk, NY, USA).

## Figures and Tables

**Figure 1 toxins-11-00349-f001:**
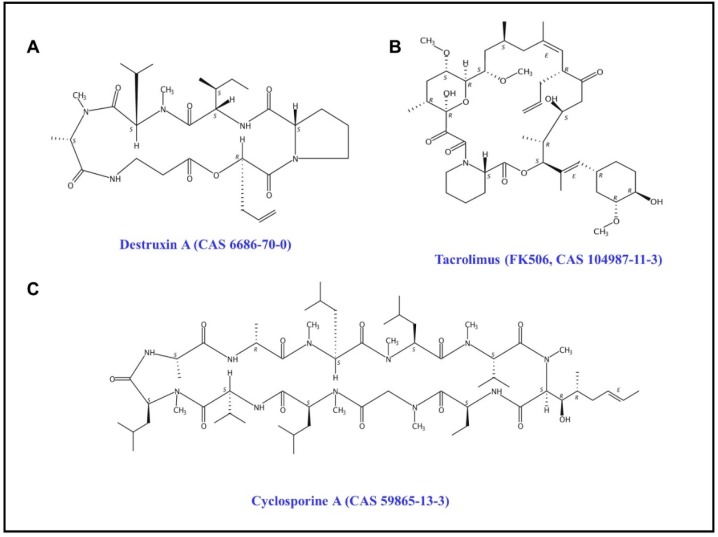
The structure of Destruxin A, tacrolimus (FK506), and cyclosporine A (CsA).

**Figure 2 toxins-11-00349-f002:**
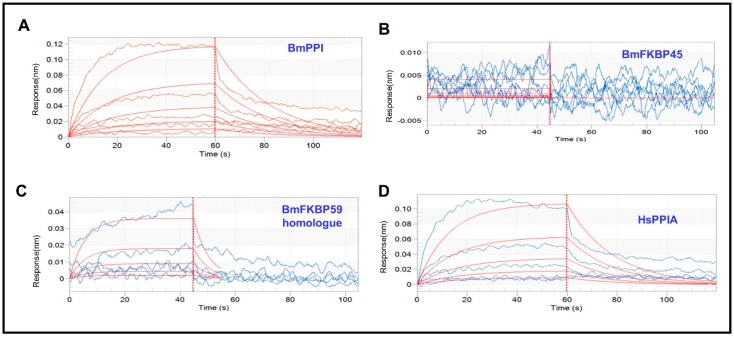
Results of affinity assay of between Destruxin A (DA) with peptidyl–prolyl cis–trans isomerase (BmPPI), FK506 binding-protein 45 (BmFKBP45), BmFKBP59 homologue and human PPI protein (HsPPIA) in bio-layer interferometry (BLI). (**A**–**D**) Data analysis of BmPPI, BmFKBP45, BmFKBP59 homologue, and HsPPIA, respectively. DA with BmPPI and HsPPIA exhibited K_D_ values of 1.98 and 2.22 × 10^−3^ M, but DA showed no interaction with BmFKBP45 and BmFKBP59 homologue.

**Figure 3 toxins-11-00349-f003:**
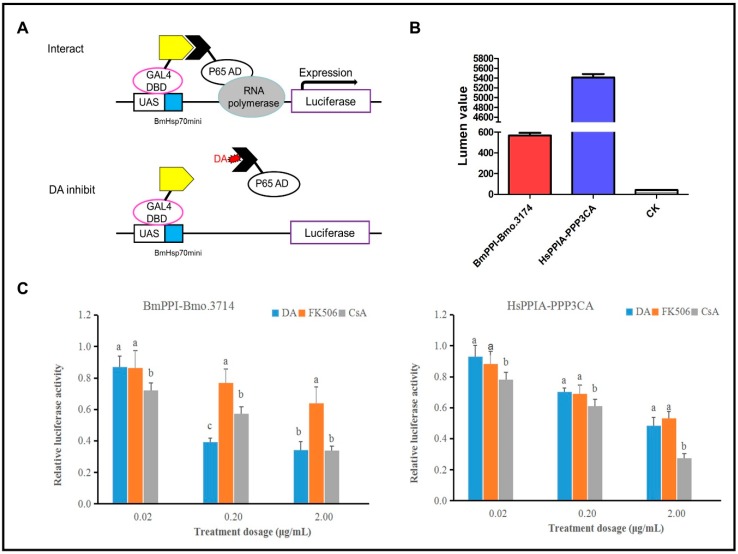
Different concentration of DA, CsA, and FK506 influences the interaction of BmPPI and HsPPIA with its interacting proteins [20,21]. (**A**) The schematic representation of the principle of insect two-hybrid (I2H). If the target protein interacts with its interacting protein, luciferase is transcriptionally activated. If a drug inhibits this interaction, luciferase cannot be expressed. (**B**) Assessing interaction between pairs. BmPPI–Bmo.3714: Sf-9 cell co-transinfected by PIE–BmPPI–AD, PIE–Bmo.3174–DBD, and PIE–Luc vectors. HsPPIA–PPP3CA: Sf-9 cell co-transinfected by PIE–HsPPIA–DBD, PIE–HsPPP3CA–AD, and PIE–Luc. CK: Sf-9 cell. (**C**) Luciferase activities of different treatment dosages of DA-, CsA-, and FK506-treated Sf9 cell line after co-transfected PIE–BmPPI–AD, PIE–Bmo.3174–DBD; PIE–HsPPIA–DBD, and PIE–HsPPP3CA–AD. The different letters on the columns indicate the significant difference (*p* < 0.05) by DMRT (Duncan’s multiple range test).

**Table 1 toxins-11-00349-t001:** The molecular interaction kinetic data of BLI assay.

Protein	DA Conc. (μM)	Response	K_D_ (M) ^1^	K_ON_ (1/Ms) ^2^	K_DIS_ (1/s) ^3^
BmPPI	62.5	0.0045	1.98 × 10^−3^	2.38 × 10^−1^	4.70 × 10^−2^
	125	0.0153			
	250	0.0261			
	500	0.0547			
	1000	0.1180			
BmFKBP45	15.6	0.0057	8.01 × 10^−3^	5.68 × 10	4.55 × 10
	31.3	0.0066			
	62.5	0.0021			
	125	0.0015			
	250	−0.0001			
	500	−0.0001			
	1000	0.0058			
BmFKBP59 homologue	62.5	0.002	1.25 × 10^−1^	1.45	0.181
	125	0.0083			
	250	0.0073			
	500	0.0159			
	1000	0.0415			
HsPPIA	62.5	0.0060	2.22 × 10^−3^	2.46 × 10^−1^	5.46 × 10^−2^
	125	0.0077			
	250	0.0247			
	500	0.0530			
	1000	0.1035			

^1^ K_D_: affinity constant. ^2^ K_ON_: association rate constant. ^3^ K_DIS_: dissociation rate constant.

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
