# Peer review of "Effects of Destruxin A on Silkworm’s Immunophilins"

_toxins, 2019, doi:10.3390/toxins11060349_

Round 1
Reviewer 1 Report
Review on:
„Effects of Destruxin A on Silkworm´s Immunophilins”
Three immunophilins (BmPPI;BmFKP45; BmFKBP59) were identified in DART assay as possible interaction partners of DA. As these experiments are not part of the manuscript and not published, essential information is missing.
In the abstract the authors stated that the immunophilins were isolated from Bm12 cell line (Line 6).
In Line 61 the authors stated that the immunophilins used for BLI assay were expressed in E.coli. The authors stated in Line 169 “All proteins were prepared by expression in the E.coli. These were tagged with His-tag and purified by nickel affinity chromatography. “This statement is all what I can find in the manuscript regarding the method and the properties of these eukaryotic proteins expressed in a prokaryotic organism.
The authors used serval abbreviations without explanation.
The organization of the manuscript is confusing and the reader is unable to follow the logic of the experiments.
Despite the complex experimental approaches the section “Materials and methods” is only 25 lines long. I doubt that anybody could reproduce the presented experiments with such minimal information.
Line 61ff: kD (M)??
Line 172: “Serial not gradient dilution…”?
Line 85: PIE-BmPPI-AD ?
Line 112: DMRT
...
Author Response
Dear reviewer,
Please refer to the attachment for detailed response.
Thanks.

Reviewer 2 Report
Description of weak interaction in living organism is a great challenge. As a part of these research, understanding the nature of interactions between proteins and bioactive molecules is in focus. In this manuscript Destruxin A binding to immunophilins have been investigated using Bio-layer interferometry assay.
1. Although bio-layer interferometry assay is a very useful technology for measuring biomolecular interactions, the binding process must be verified by another independent measurements. Several analytical methods (e.g. UV-vis, fluorescence, affinity chromatography, equilibrium dialysis, X-ray crystallography, circular dichroism, IR and Raman spectroscopy, NMR, surface plasma resonance etc.) are capable to investigate the protein-ligand interactions. In my opinion, one method is certainly available for authors to justify the Destruxin A binding to immunophilins found by the bio-layer interferometry method. The results will be interpreted in two independent measurements.
2. The figures and tables are incomprehensible because of the low resolution. Furthermore, their form does not meet the expectations of the journal.
3. The details of the mentioned statistics (“Figure 3. … The different letters on the columns indicate the significant difference (p < 0.05) by DMRT.”) must be inserted into Section 5. Materials and Methods.
Author Response

(The authors gave the same response as above.)

Round 2
Reviewer 1 Report
Dear authors,
despite your response to the critcal points some points remain:
Abbbreviation kD is usually reserved for "kilo Dalton" (kD) as an old, but still valid unit of molecular mass. The dissociation constant is K sub D.
Line 102, B: Legende at y-axis!
Line 187: ..of the plasmid-...
Line 188ff: ..2 µL lipofectamine 2000
Literatur:
Line 208ff: ...MHJTOJotSoT..??
Please control carefully the format of all citations!
Author Response

(The authors gave the same response as above.)

Reviewer 2 Report
The authors did not answer my questions and did not make the requested changes (see e.g. Figures or Tables).
Author Response

(The authors gave the same response as above.)

Round 3
Reviewer 2 Report
I have not any other new questions or suggestions.